# Studies on the Effects of Hypercholesterolemia on Mouse Ophthalmic Artery Reactivity

**DOI:** 10.3390/diseases11040124

**Published:** 2023-09-22

**Authors:** Francesco Buonfiglio, Ning Xia, Can Yüksel, Caroline Manicam, Subao Jiang, Jenia Kouchek Zadeh, Aytan Musayeva, Eva Elksne, Norbert Pfeiffer, Andreas Patzak, Huige Li, Adrian Gericke

**Affiliations:** 1Department of Ophthalmology, University Medical Center, Johannes Gutenberg University Mainz, Langenbeckstr. 1, 55131 Mainz, Germany; 2Department of Pharmacology, University Medical Center, Johannes Gutenberg University Mainz, Langenbeckstr. 1, 55131 Mainz, Germany; 3Institute of Translational Physiology, Charité—Universitätsmedizin Berlin, Charitéplatz 1, 10117 Berlin, Germany

**Keywords:** aorta, apolipoprotein E, hypercholesterolemia, ophthalmic artery, oxidative stress

## Abstract

Atherogenic lipoproteins may impair vascular reactivity, leading to tissue damage in various organs, including the eye. This study aimed to investigate whether ophthalmic artery reactivity is affected in mice lacking the apolipoprotein E gene (ApoE−/−), a model for hypercholesterolemia and atherosclerosis. Twelve-month-old male ApoE−/− mice and age-matched wild-type controls were used to assess vascular reactivity using videomicroscopy. Moreover, the vascular mechanics, lipid content, levels of reactive oxygen species (ROS), and expression of pro-oxidant redox enzymes and the lectin-like oxidized low-density lipoprotein receptor-1 (LOX-1) were determined in vascular tissue. Unlike the aorta, the ophthalmic artery of ApoE−/− mice developed no signs of endothelial dysfunction and no signs of excessive lipid deposition. Remarkably, the levels of ROS, nicotinamide adenine dinucleotide phosphate oxidase 1 (NOX1), NOX2, NOX4, and LOX-1 were increased in the aorta but not in the ophthalmic artery of ApoE−/− mice. Our findings suggest that ApoE−/− mice develop endothelial dysfunction in the aorta by increased oxidative stress via the involvement of LOX-1, NOX1, and NOX2, whereas NOX4 may participate in media remodeling. In contrast, the ophthalmic artery appears to be resistant to chronic apolipoprotein E deficiency. A lack of LOX-1 expression/overexpression in response to increased oxidized low-density lipoprotein levels may be a possible mechanism of action.

## 1. Introduction

Hypercholesterolemia stands as one of the primary risk factors for atherosclerosis and other cardiovascular diseases [1,2,3,4]. Notably, this condition is positively associated with retinal artery and vein occlusion [5,6,7]. Furthermore, heightened serum cholesterol levels have been associated with diminished retinal vascular hyperemic responses in humans when subjected to flicker light stimulation [8,9]. Despite these observations, the impact of hypercholesterolemia on vascular signaling pathways in the ocular circulation remains poorly understood. Studies conducted in different vascular beds have shed light on pivotal molecular processes that contribute to vascular damage induced by hypercholesterolemia, often exacerbated by the presence of reactive oxygen species (ROS). These processes encompass the oxidation of lipoproteins, activation of endothelial cells, and infiltration of macrophages into the vascular wall [10]. Within the vascular wall, several enzymes can be identified as sources of ROS. These include the reduced form of nicotinamide adenine dinucleotide phosphate (NADPH) oxidase (NOX), xanthine oxidase (XO), mitochondrial electron transport chain, and malfunctioning endothelial nitric oxide synthase (eNOS) [10,11,12]. The primary objective of this study was to investigate the hypothesis that chronic hypercholesterolemia causes endothelial dysfunction in the ophthalmic artery, which is the first branch of the internal carotid artery supplying blood to the eye and other orbital structures. Additionally, we aimed to assess whether intracellular signaling pathways involved in vasodilation are affected by hypercholesterolemia. To conduct these investigations, we utilized mice lacking the apolipoprotein E gene (ApoE−/−), an established mouse model known to develop severe spontaneous hypercholesterolemia and atherosclerotic lesions in various blood vessels, including the aorta, similar to those found in humans [13,14,15].

## 2. Material and Methods

### 2.1. Animals

All experimental procedures were conducted in compliance with ethical standards and received prior approval from the Animal Care Committee of Rhineland-Palatinate, Germany, under approval number 23 177-07/G 13-1-034. The experiments strictly adhered to the guidelines outlined in the EU Directive 2010/63/EU for animal experiments.

Mice deficient in the gene encoding apolipoprotein E (ApoE−/−) and wild-type controls (C57BL/6J) were procured from The Jackson Laboratory, located in Bar Harbor, ME, USA. Male mice, aged 12 months, were selected for the experiments. These mice were maintained on a standard rodent chow (Altromin, Lage, Germany). The mice were housed in a carefully controlled environment, featuring a 12 h light/dark cycle, a constant temperature of 22 ± 2 °C, humidity levels maintained at 55 ± 10%, and provided with unrestricted access to both food and tap water.

### 2.2. Measurement of Blood Pressure and Total Serum Cholesterol

Blood pressure measurements were conducted in conscious restrained mice using a computerized tail-cuff system (CODA® Monitor, Kent Scientific, Torrington, CT, USA). To familiarize the mice with the procedure, they underwent a two-day training period. During measurements, mice were placed in restraint tubes to minimize movement and positioned on a warming platform (maintained at 32–35 °C) to ensure comfort. Before starting the actual experiment, the tails of the mice were gently cuffed, and an acclimatization period of 5 min was allowed to let the mice adjust to the setup. Each measurement session comprised 20 cycles, with the initial 5 cycles designated for acclimatization and excluded from the analysis. The average of the subsequent 15 cycles was recorded as the blood pressure reading for each mouse. Following the experiment, mice were euthanized by CO_2_ exposure. Blood samples were collected from the heart, and the serum total cholesterol levels were determined using the scil Reflovet® Plus analyzer (scil animal care company GmbH, Viernheim, Germany).

### 2.3. Pharmacological Studies

Pharmacological studies were performed in the aorta, which is known for developing endothelial dysfunction and morphologic changes in ApoE−/− mice and therefore served as a positive control, and in the ophthalmic artery [13,14,15]. Mice were euthanized by inhalation of CO_2_. Subsequently, the abdominal aorta and the ophthalmic artery were carefully dissected and placed in cold Krebs–Henseleit buffer with the following ionic composition (in mM): 118.3 NaCl, 25.0 NaHCO_3_, 11.0 glucose, 4.7 KCl, 2.5 CaCl_2_, 1.2 MgSO_4_, and 1.2 KH_2_PO_4_ (Carl Roth GmbH, Karlsruhe, Germany). After cleaning the segments of the abdominal aorta and the ophthalmic artery from surrounding tissue, they were transferred into a perfusion chamber and cannulated and sutured onto micropipettes as previously described [15,16,17]. The segments were pressurized using the pipettes to 80 mm Hg for the aorta and 50 mm Hg for the ophthalmic artery by utilizing two reservoirs filled with Krebs–Henseleit buffer. The vessels were then visualized using a video camera attached to an inverted microscope. The perfusion chamber was continuously supplied with carbonated and oxygenated Krebs–Henseleit buffer at 37 °C and maintained at pH 7.4. An equilibration period of 45 min preceded the commencement of functional studies. The suitability of the blood vessels for experiments was ensured by confirming a minimum of 20% constriction from the resting luminal vessel diameter after perfusion with high KCl solution (100 mM) for the aorta or 50% constriction for the ophthalmic artery.

Next, concentration–response curves were gained for phenylephrine (10^−8^–10^−3^ M, Sigma-Aldrich Chemie GmbH, Munich, Germany), an α_1_-adrenoceptor agonist. Concentration–response curves were also obtained for two vasodilators: nitroprusside (10^−9^–10^−4^ M, Sigma-Aldrich), an endothelium-independent vasodilator, and acetylcholine (10^−9^–10^−4^ M, Sigma-Aldrich), an endothelium-dependent vasodilator, in the same vessel segments preconstricted with phenylephrine to 70–50% of their initial diameter.

In a subsequent set of experiments, we applied various inhibitors to investigate the pathways involved in endothelium-dependent vasodilation in the ophthalmic artery. After obtaining a concentration–response curve for acetylcholine (10^−9^–10^−4^ M) in arteries preconstricted with phenylephrine, the vessels underwent a 10 min washout period. Subsequently, they were incubated for 30 min with the respective blocker. The vessel was then preconstricted again to a similar level as before, and another concentration–response curve for acetylcholine (10^−9^–10^−4^ M) was obtained.

To investigate the involvement of different NOS isoforms in endothelium-dependent vasodilation responses, we subjected the vessels to incubation with specific inhibitors. These inhibitors included Nω-nitro-L-arginine methyl ester (L-NAME, 10^−4^ M, Sigma-Aldrich), a non-isoform-selective NOS inhibitor, 1-[2-(trifluoromethyl)phenyl]imidazole (TRIM, 10^−4^ M, Sigma-Aldrich), a selective inhibitor of neuronal NOS (nNOS), and aminoguanidine (3 × 10^−4^ M, Sigma-Aldrich), which selectively inhibits inducible NOS (iNOS). Given the absence of highly specific inhibitors for eNOS currently available, the contribution of eNOS to vasodilation responses had to be inferred from experiments involving nNOS, iNOS, and the non-isoform-selective NOS inhibitor.

To assess the contribution of cyclooxygenase (COX) metabolites to endothelium-dependent vasodilation, reactivity to acetylcholine before and after incubation with indomethacin (10^−5^ M, Sigma-Aldrich), a non-isoform-selective inhibitor of COX-1 and COX-2, was tested. Moreover, the effects of combined blockade of COX and NOS by simultaneously incubating with L-NAME (10^−4^ M) and indomethacin (10^−5^ M) were tested, to exclude the possibility that both signaling pathways compensate for each other. To assess the contribution of endothelium-derived hyperpolarizing factors (EDHFs) to acetylcholine-induced vasodilation, responsiveness of the ophthalmic artery to acetylcholine was tested in the presence of L-NAME and indomethacin before and after incubation with KCl (30 mM), which prevents hyperpolarization and thus blocks the EDHF-dependent portion of vasodilation responses.

To explore potential EDHF pathways involved, we assessed the ophthalmic artery’s reactivity to acetylcholine under various conditions. First, reactivity was tested in the presence of L-NAME and indomethacin. Subsequently, we conducted experiments after incubation with specific compounds, such as 17-octadecynoic acid (17-ODYA, 10^−4^ M, Tocris Bioscience, Bristol, UK). This compound serves as a suicide substrate inhibitor, affecting both ω-hydroxylation and epoxygenation of arachidonic acid via the cytochrome P450 monoxygenase (CYP450) pathway. We also incubated with baicalein (10^−5^ M, Sigma-Aldrich), which is a specific inhibitor of 12/15-lipoxygenase. Moreover, we applied 18α-glycyrrhetinic acid (18α-GA, 3 × 10^−5^ M, Sigma-Aldrich), which acts as a gap junction blocker. These pathways were previously shown to be involved in EDFH-dependent ophthalmic artery responses of wild-type mice of the C57Bl/6 J background [18].

### 2.4. Vascular Mechanics

Chronic hypertension and hypercholesteremia have been reported as triggers of a remodeling in the blood vessel walls [19,20]. These conditions can lead to an alteration of the vascular mechanics, more specifically of the stress–strain relation, as previously described in mouse arteries [21]. For studies of vascular mechanics, the ophthalmic artery smooth muscle was deactivated by incubation with Krebs buffer containing 67 mmol/L etheylenediaminetetraacetic acid (EDTA) for 20 min, which produces complete deactivation of smooth muscle [22]. Then, pressure–diameter relationships were obtained in deactivated ophthalmic arteries between luminal pressures of 80 and 0 mmHg in steps of 10 mmHg, similar to previous reports [23]. At each pressure level, the artery segment was allowed to equilibrate for 1 min before internal and external diameters were measured. Circumferential stress (σ) was calculated from intravascular pressure (P), internal diameter (Di), and wall thickness (WT): σ = (P × Di)/(2 × WT). Circumferential strain (ε) was calculated by the following formula: ε = (ΔD/D₀)/D_0_. D_0_ represents the original diameter, defined as the internal diameter at a pressure of 0 mmHg.

### 2.5. Oil Red O Staining

Oil Red O (ORO) staining was used to assess lipid content in cross-sections of the aorta and the ophthalmic artery from ApoE−/− and wild-type mice. Cryosections (10 µm) were air-dried, fixed with 4% paraformaldehyde for 5 min, rinsed with running tap water for 1 min, and dehydrated with 60% isopropanol. Next, the tissue sections were stained with freshly prepared ORO solution (Carl Roth GmbH, Karlsruhe, Germany; 0.5 g in 100 mL isopropanol and diluted with distilled water at a ratio of 3:2) for 15 min and then rinsed with 60% isopropanol, stained with hematoxylin (5 dips), rinsed with distilled water, and finally mounted with glycerin jelly and cover-slipped. For measurements of staining intensity, the hematoxylin step was omitted. Staining was measured by a blinded evaluator using ImageJ software, version 1.53m (NIH, http://rsb.info.nih.gov/ij/ (accessed on 10 October 2021)). Phthalmic arteries of both mouse groups contain dark, melanin-enriched deposits along some fragments of their vascular wall, especially in the adventitia but also partially in the media, these dark fields were excluded from densitometric analysis.

### 2.6. Quantification of ROS

Superoxide formation was evaluated in cryosections of both the aorta and the ophthalmic artery using the oxidative fluorescent dye dihydroethidium (DHE), following previously established procedures [24]. Upon sacrificing the mice and isolating the aorta and ophthalmic artery, we prepared frozen cross-sections with a thickness of 10 µm. Following thawing, the tissue sections were promptly exposed to 10 µmol/l DHE for a duration of 30 min. Subsequently, photographs of the vascular cross-sections were captured using a fluorescent microscope, employing an excitation wavelength of 520 nm and an emission wavelength of 610 nm. To quantify the staining intensity, a blinded evaluator utilized ImageJ software (NIH, http://rsb.info.nih.gov/ij/ (accessed on 10 October 2021)).

### 2.7. Quantification of Redox Genes by Real-Time PCR

We quantified the messenger RNA levels of pro-oxidative redox enzymes, including NOX1, NOX2, NOX4, and xanthine dehydrogenase (XDH), in both the aorta and the ophthalmic artery from both mouse genotypes using real-time PCR. Under normal physiological conditions, XDH functions predominantly as a dehydrogenase. However, it is important to note that various stimuli, such as hypoxia and inflammation, can trigger the conversion of XDH into the superoxide-producing enzyme known as xanthine oxidase (XO). This enzymatic conversion to XO has been associated with the promotion of endothelial dysfunction and the development of atherosclerosis [25].

Immediately after animals’ death, blood vessels were rapidly excised and placed in cold phosphate-buffered solution (PBS, Invitrogen, Karlsruhe, Germany). Under a stereo microscope, vessels were cleaned from surrounding tissue and blood and then cut into small pieces. Next, the tissue samples were transferred into a 1.5 mL tube and rapidly frozen in liquid nitrogen. Following sample collection, we proceeded to homogenize them using the FastPrep system (MP Biomedicals, Illkirch, France), and total RNA extraction was carried out utilizing the RNeasy Kit (QIAGEN, Hilden, Germany) in accordance with the manufacturer’s instructions. Subsequently, the mRNA was reverse-transcribed using M-MLV reverse transcriptase and random hexamers (Promega, Mannheim, Germany). Quantitative PCR analysis was performed employing the ViiA 7 system (Applied Biosystems, Darmstadt, Germany), with SYBR green utilized for DNA detection. Relative mRNA levels of the target genes were quantified through the comparative threshold (CT) method and normalized to β-actin (*ACTB*). The expression of messenger RNA is presented as the fold change in ApoE−/− mice compared to wild-type mice. The specific PCR primer sequences can be found in Table 1.

### 2.8. Immunostainings

To visualize and evaluate the expression levels of pro-oxidative enzymes (NOX1, NOX2, NOX4, and XDH/XO) and LOX-1, we conducted the following procedures on aortas and ophthalmic arteries obtained from both wild-type and ApoE−/− mice: First, aortas and ophthalmic arteries were meticulously dissected and then immersed in 4% paraformaldehyde overnight. Subsequently, they were embedded into paraffin blocks, and cross-sections of the vasculature, measuring 7 µm in thickness, were generated. These cross-sections were subjected to immunostaining using rabbit antibodies specific to NOX1, NOX2, NOX4, XDH/XO, or LOX-1 (see Table 2 for specifications). As a secondary antibody, a Rhodamine Red-X-conjugated goat anti-rabbit polyclonal antibody (Dianova GmbH, Hamburg, Germany; catalog number: 111-295-003) was applied at a dilution of 1:200, with an incubation period of 1 h at room temperature. Negative control sections were treated with blocking media and the secondary antibody. Finally, the prepared slides were mounted using VECTASHIELD® Mounting Medium with DAPI (BIOZOL Diagnostica Vertrieb GmbH, Eching, Germany) and cover-slipped.

### 2.9. Statistics

Responses to the vasoconstrictor, phenylephrine, are shown as relative change in luminal vessel diameter from resting diameter, while responses to the vasodilators, acetylcholine and nitroprusside, are expressed as relative change in luminal diameter from preconstricted diameter. Two-way ANOVA for repeated measurements was used to compare concentration–response curves. Blood pressure levels, total serum cholesterol, staining intensity, and mRNA expression levels were compared using the unpaired *t* test. The significance level was 0.05, and n represents the number of mice.

## 3. Results

### 3.1. Blood Pressure and Total Serum Cholesterol

Systolic, diastolic, and mean blood pressure levels did not differ between ApoE−/− and wild-type mice (Table 3). However, total serum cholesterol was elevated by more than threefold in ApoE−/− mice compared to wild-type mice (*p* < 0.0001, n = 8 per genotype, Table 3), indicative of pronounced hypercholesterolemia.

### 3.2. Vascular Responses to Pharmacological Agents

Phenylephrine (10^−8^–10^−3^ M) elicited concentration-dependent vasoconstriction of the aorta that was similar in ApoE−/− and wild-type mice. The maximal reduction in luminal diameter was 59 ± 8.9% and 57 ± 6.5% in ApoE−/− and wild-type mice at 10^−3^ M (Figure 1A). In aortas preconstricted with phenylephrine, the endothelium-independent vasodilator, nitroprusside (10^−9^–10^−4^ M), produced concentration-dependent vasodilation, which did not differ between aortas from ApoE−/− and wild-type mice. The increase in luminal diameter to 10^−4^ M nitroprusside was 54 ± 11% and 63 ± 14% in ApoE−/− and wild-type mice, respectively (Figure 1B). In contrast, the responses of aortas to the endothelium-dependent vasodilator, acetylcholine (10^−9^–10^−4^ M), were markedly reduced in ApoE−/− mice. For example, the increase in luminal diameter to 10^−4^ M acetylcholine was 26 ± 11% and 67 ± 9.7% (*p* < 0.01) in ApoE−/− and wild-type mice, respectively, indicative of endothelial dysfunction in the ApoE−/− mouse aorta (Figure 1C). In the ophthalmic artery, phenylephrine (10^−8^–10^−3^ M) concentration-dependently elicited vasoconstrictor responses that were similar in ApoE−/− and wild-type mice. The maximal reduction in luminal diameter was 55 ± 8.0% and 51 ± 5.4% at 10^−3^ M in ApoE−/− and wild-type mice (Figure 1D). The endothelium-independent vasodilator, nitroprusside (10^−9^–10^−4^ M), elicited concentration-dependent vasodilation, which also did not differ between the ophthalmic artery from ApoE−/− and wild-type mice. The increase in luminal diameter in response to 10^−4^ M nitroprusside was 68 ± 13% and 78 ± 11% in ApoE−/− and wild-type mice, respectively (Figure 1E). Of note, ophthalmic artery responses to the endothelium-dependent vasodilator, acetylcholine (10^−9^–10^−4^ M), were similar in ApoE−/− and wild-type mice. The diameter increase in response to 10^−4^ M of acetylcholine was 74 ± 6.0% and 76 ± 8.5% in ApoE−/− and wild-type mice, respectively (Figure 1F).

Previous studies have shown that small arteries, including the ophthalmic artery, possess the ability to compensate for the blockade or lack of a vasodilatory pathway by activating alternative signaling pathways that retain vasodilation responses [26,27]. To test whether endothelial signaling pathways were changed in the ophthalmic artery of ApoE−/− mice, we performed additional experiments using various pharmacological blockers. After exposure to the non-isoform-selective NOS blocker, L-NAME, acetylcholine-induced responses were reduced to a similar degree in ApoE−/− and wild-type mice (Figure 2A). Neither the selective nNOS inhibitor, TRIM (Figure 2B), nor the selective iNOS blocker, aminoguanidine (Figure 2C), had any effect on cholinergic ophthalmic artery responses in both genotypes. Incubation with indomethacin alone did not affect ophthalmic artery responses to acetylcholine (Figure 2D), and combined blockade with indomethacin and L-NAME reduced cholinergic responses in both genotypes similar to incubation with L-NAME alone (Figure 2E), indicating that NOS and COX did not compensate for each other. After the addition of 30 mM KCl to ophthalmic arteries preincubated with indomethacin and L-NAME, acetylcholine-induced vasodilation was negligible in both mouse genotypes (Figure 2F), indicative of EDHF involvement. To further elucidate the role of EDHF in this vascular bed of ApoE−/− mice, several blockers and inhibitors targeting previously identified signaling pathways in the mouse ophthalmic artery were employed [18]. First, incubation with 17-ODYA, a blocker of CYP450, similarly reduced EDHF-mediated responses to acetylcholine in ApoE−/− and wild-type mice as shown in Figure 2G. Also, baicalein blunted EDHF-mediated responses to a similar extent in both mouse genotypes (Figure 2H). Likewise, 18α-GA similarly reduced EDHF-mediated vasodilation in both genotypes (Figure 2I).

### 3.3. Lipid Content and Vascular Mechanics

Unlike in wild-type mice, the aorta of ApoE−/− mice was rich in lipids as visualized by ORO. Especially, lipid infiltration was visible in the endothelium, which was rich in large lipid drops (Figure 3A). In the media, spots with abundant lipid content were frequently seen in the aorta of ApoE−/− mice (Figure 3A). They were especially dense in areas with atherosclerotic plaques. In the ophthalmic artery of both wild-type and ApoE−/− mice, small lipid drops were visible in the media but rarely in the endothelium. Remarkably, no differences in lipid content were observed between ophthalmic arteries from wild-type and ApoE−/− mice (Figure 3B).

To further test whether vascular remodeling, reflected in altered vascular mechanics, occurred in the ophthalmic artery of ApoE−/− mice, the relation of internal and external vascular diameters to the luminal pressure, as well as the stress–strain relation, were assessed. However, pressure–diameter relations and stress-strain curves were similar in wild-type and ApoE−/− mice, suggesting that no significant remodeling occurred in the ophthalmic artery vascular wall of ApoE−/− mice (Figure 3C).

### 3.4. ROS Levels in the Vascular Wall

Staining of aortic cross-sections with DHE unveiled a significant elevation in fluorescent intensity within the vascular wall of ApoE−/− mice, indicative of a heightened concentration of ROS (*p* < 0.01, ApoE−/− versus wild-type mice; Figure 4A). In contrast, no discernible differences in staining intensity were observed when comparing ophthalmic arteries from ApoE−/− and wild-type mice (Figure 4B).

### 3.5. Vascular Redox Gene Expression

Among the three NOX isoforms (NOX1, NOX2, and NOX4) and the XO that were previously shown to be involved in lipoprotein-mediated pathological conditions of the aorta, the expression level of NOX4 mRNA was found to be about 15 times higher in ApoE−/− mice compared to wild-type mice (*p* < 0.001; Figure 5A). In contrast, no significant differences in mRNA expression levels of the four pro-oxidative redox genes were found between ophthalmic arteries from ApoE−/− and wild-type mice, as shown in Figure 5B. To assess the expression of the redox enzymes on the protein level and to localize them within the vascular wall, immunostainings were conducted. Interestingly, a pronounced expression of NOX1 and NOX2 was detected in the vascular endothelium of the aorta from ApoE−/− mice, whereas NOX4 expression was not increased in the endothelium but in the smooth muscle layer. Remarkably, no marked differences in expression of the pro-oxidant redox enzymes were detected between the ophthalmic artery of wild-type and ApoE−/− mice.

### 3.6. Redox Enzyme Expression in the Vascular Wall

While the aorta of wild-type mice had only negligible immunoreactivity for the prooxidative enzyme, NOX1, higher immunoreactivity was found in the endothelial layer of the ApoE−/− mouse aorta (Figure 6A). A similar picture was found for NOX2, although immunoreactivity was even more pronounced than for NOX1 (Figure 6B). Intriguingly, only negligible NOX4 immunoreactivity was found in the endothelium of both the wild-type and the ApoE−/− mouse aorta (Figure 6C). However, in the aorta of ApoE−/− mice, abundant immunoreactivity for NOX4 was found in the media layer, especially in the areas with plaque formation (Figure 6C). For XDH/XO, no immunoreactivity was found in the aorta of both mouse genotypes (Figure 6D).

Notably, only negligible immunoreactivity was found in the ophthalmic artery endothelium and media of both genotypes for NOX1 (Figure 7A), NOX2 (Figure 7B), NOX4 (Figure 7C), and XDH/XO (Figure 7D).

### 3.7. Expression of LOX-1 in the Vascular Wall

Immunoreactivity to LOX-1 was very weak in the aorta from wild-type mice. In contrast, LOX-1 immunoreactivity was pronounced in the endothelium and in the smooth muscle layer of the aorta. (Figure 8A). Remarkably, immunoreactivity was very faint in the endothelium of the ophthalmic artery from both ApoE−/− and wild-type mice. Although the ophthalmic artery smooth muscle layer displayed remarkable immunoreactivity to LOX-1, there was no difference in fluorescent intensity between ApoE−/− and wild-type mice (Figure 8B).

## 4. Discussion

There are several new findings in this study. First, aortas from 12-month-old ApoE−/− mice displayed impaired vasodilation responses to acetylcholine and were characterized by excessive lipid storage in the vascular wall, especially the endothelium. Additionally, aortas displayed elevated ROS, NOX1, NOX2, and NOX4 expression levels and increased LOX-1 immunoreactivity in the endothelium and smooth muscle, indicative of oxidative stress-mediated vascular damage. In contrast, the ophthalmic arteries from ApoE−/− mice had preserved vasoconstriction and vasodilation responses, normal vascular mechanics, no signs of histological changes, normal ROS and pro-oxidative enzyme levels, and no signs of LOX-1 upregulation in the vascular wall. Likewise, the contribution of signaling pathways to acetylcholine-induced vasodilation did not differ between the ophthalmic artery of ApoE−/− and wild-type mice, suggesting that no compensatory pathways have been activated in ApoE−/− mice to retain endothelium-dependent vasodilation.

During hypercholesterolemia, oxidized low-density lipoproteins (Ox-LDL) may trigger the expression of pro-oxidative enzymes in the vascular wall that contribute to the generation of ROS, which at high concentrations, elicit endothelial dysfunction in part by affecting eNOS function and by inactivation of nitric oxide [28,29,30,31]. Prior studies involving ApoE−/− mice have detected markedly elevated plasma Ox-LDL levels under standard mouse chow conditions and even more extensive under atherogenic diet conditions [32,33,34,35]. This increase in Ox-LDL levels has been implicated in the progression of atherosclerotic lesions [33].

Impaired responses to endothelium-dependent vasodilators have been reported in both large and small arteries of hypercholesterolemic patients [36,37,38,39,40]. Of note, in the human retina, vascular hyperemic responses to flicker light stimulation were reduced with increasing serum cholesterol levels even within the physiological range [8,9]. Likewise, ApoE−/− mice were reported to develop endothelial dysfunction in both large and small blood vessels, including retinal arterioles [41,42,43,44]. 

However, there were also reports on experiments in small arteries of hypercholesterolemic animal models and humans that did not develop an impairment of endothelial function. For example, in isolated mesenteric arteries from ApoE−/− mice, endothelium-dependent relaxations to acetylcholine remained intact or were even enhanced [27]. In another model of dyslipidemia, EDHF-mediated reactivity of gracilis muscle arteries in response to acetylcholine was also shown to be enhanced [45]. Similarly, in cremaster muscle arterioles of ApoE−/− and LDL receptor-deficient mice, endothelium-dependent vasodilation was retained [46]. In human gastroepiploic arteries, hypercholesterolemia significantly reduced responses in large, but not in small vessels [47].

However, why endothelium-dependent vasodilation of some small arteries is barely affected by hypercholesterolemia is unknown at present. Possible explanations are reduced uptake of Ox-LDLs by endothelial cells, resistance of intracellular pro-oxidative signaling pathways to activation by Ox-LDLs, increased intracellular antioxidant enzyme activity, or alternative signaling pathways that are less susceptible to the effects of hypercholesterolemia. For example, large conductance, Ca^2+^-activated K^+^ channels (BK_Ca_) have been proposed to rescue EDHF-dependent vasodilation responses in mesenteric arteries of ApoE−/− mice [27]. In another study on gracilis muscle arteries from dyslipidemic mice, a compensatory role for a CYP450 metabolite of arachidonic acid was proposed to preserve endothelial function [45]. Specifically, the activity of the CYP450 generates epoxyecosatrienoic acids (EETs), facilitating vasodilation through gap junction communication [18]. In this context, the 12/15 lipoxygenase (12/15 LOX) enzyme also plays a pivotal role in physiological vasorelaxation by producing 12-hydroxyeicosatetraenoic acid (12-HETE) [48]. These arachidonic acid metabolites trigger an increase in intracellular potassium (K⁺) concentration within the vascular smooth muscle layer via gap junctions, ultimately enabling normal vasodilation [18].

Previous studies from our laboratory revealed that in the mouse ophthalmic artery, vasodilation responses to acetylcholine are mediated by M_3_ receptors and several endothelium-dependent mechanisms involving eNOS and lipoxygenase- and CYP450-derived metabolites, inwardly rectifying potassium (Kir) channels and gap junctions [16,18,49,50]. In contrast, during chronic genetic eNOS deficiency or angiotensin II exposure, which induces oxidative stress and eNOS dysfunction, endothelium-dependent vasodilation was predominantly mediated by EDHF-dependent signaling pathways, suggesting that the mouse ophthalmic artery is equipped with several signaling mechanisms that may help to retain endothelium-dependent vasodilation when the primary signaling pathway is dysfunctional [26,48]. In the present study, eNOS contributed to a similar extent to endothelium-dependent vasodilation in the ophthalmic artery of both mouse genotypes indicating that eNOS function was normal in the ophthalmic artery of ApoE−/− mice. Moreover, there was no hint that alternative signaling pathways had been activated in the ophthalmic artery from ApoE−/− mice, because a blockade of nNOS, iNOS, COX, lipoxygenase, CYP450, and gap junctions had similar effects on endothelium-dependent vasodilation as in wild-type mice. All together, these findings suggest that hypercholesterolemia did not exert marked effects on endothelial function in the ophthalmic artery, which contrasts with our previously published findings in retinal arterioles, where pronounced endothelial dysfunction was observed in ApoE−/− mice [51].

We used ApoE−/− mice for our studies, because this is an established mouse model that develops spontaneous severe hypercholesterolemia, elevated vascular ROS and cytokine levels, endothelial dysfunction, and atherosclerotic lesions similar to those seen in humans in various arteries [13,41,44]. In the present study, ApoE−/− mice had markedly elevated ROS levels and a reduced endothelial function in the aorta, which is in line with previous studies [15,52]. Interestingly, in contrast to the findings in the aorta, no elevation of ROS levels was found in the ophthalmic artery.

The pro-oxidant enzymes NOX and XDH/XO were previously shown to contribute to vascular damage during hypercholesterolemia in ApoE−/− mice [53,54,55]. However, the role of individual NOX isoforms in atherosclerosis is heavily debated. While NOX1 and NOX2 were mostly shown to contribute to atherosclerotic lesions, the role of NOX4 is controversial [54,56]. Whereas some studies support the concept that NOX4 is a mediator of atherosclerosis during hypercholesterolemia, other groups have reported opposing findings [57,58,59,60,61,62].

In the present study, we found that NOX4 mRNA expression was 15-fold higher in the abdominal aorta of ApoE−/− mice compared to the wild-type controls, whereas NOX1, NOX2, and XDH/XO mRNA levels did not differ. Interestingly, NOX4 protein expression was upregulated in the media, but not in the endothelium of the ApoE−/− mouse aorta, suggesting that NOX4 did not contribute to endothelial dysfunction. Previous studies suggested a role of NOX4 in cell growth, apoptosis, senescence, and autophagy of aortic smooth muscle cells during hypercholesterolemia [58,63]. An overabundance of ROS is possibly able to directly activate TGF-β1, a pivotal mediator of fibrogenesis in the vascular smooth muscle layer [64]. This fibrogenetic regulator may in turn upregulate NOX4, creating a vicious cycle and a relevant link between oxidative stress and fibrotic processes in the cardiovascular system [58,65,66]. 

Conversely, the expression of NOX1 and NOX2 proteins exhibited upregulation specifically within the endothelium of the ApoE−/− mouse aorta, implying their involvement in the generation of ROS and subsequent endothelial dysfunction. This observation lends credence to the idea that the absence of elevated mRNA levels for these enzyme isoforms is reasonable, given the localized nature of their upregulation.

Notably, none of the tested pro-oxidant enzymes displayed elevated mRNA and protein expression levels in the ophthalmic artery from ApoE−/− mice, which is in line with the normal ROS levels and the retained endothelial signaling pathways. Since functional and morphological changes could not be found in the ophthalmic artery of ApoE−/− mice despite marked changes in the aorta, we assume that LOX-1 receptor signaling may be responsible for the difference between these two vascular beds. In a previous study on transgenic ApoE−/− mice overexpressing LOX-1, enhancement of Ox-LDL uptake was consistent with the expression level of LOX-1. Notably, these mice were characterized by elevated oxidative stress markers, increased levels of the adhesion molecules, such as the intercellular adhesion molecule 1 (ICAM-1) and the vascular cell adhesion molecule 1 (VCAM-1), pronounced macrophage infiltration, and highly increased atheroma-like lesion areas in coronary arteries [67]. An interaction of Ox-LDL and LOX-1 receptor triggers a membrane type 1 matrix metalloproteinase (MT-1-MM), activating two GTPases: the Ras homolog family member A (RhoA) and the Ras-related C3 botulinum toxin substrate 1 (Rac1) [68]. While RhoA downregulates eNOS, Rac1 induces an increased activity of NOX enzymes eventually leading to ROS formation [31,68]. Another study suggested that LOX-1 expression and its translocation into the cell membrane can be triggered by Ox-LDL itself in some tissues/blood vessels [28]. In line with these findings, we found that the expression of LOX-1 is indeed upregulated in the aorta of ApoE−/− mice. However, in the ophthalmic artery, no such evidence for LOX-1 upregulation in ApoE−/− mice could be found. Since we also did not find any evidence for oxidative stress, morphological and mechanical changes, endothelial dysfunction, or compensation of endothelial function by alternative vasodilatory signaling pathways in the ophthalmic artery, we assume that the absence of LOX-1 signaling into the cell may be a possible reason for the resistance of this vascular bed to chronic hypercholesterolemia. Based on the findings, we propose the following scheme (Figure 9).

Our findings support a previous study in humans that did not reveal any differences in retrobulbar blood flow velocities between hypercholesterolemic and control subjects although differences in finger nailfold capillary blood flow velocities have been found [69]. Both studies indicate that retrobulbar vessel function may be more resistant to increased levels of atherogenic lipoproteins than in other vascular beds.

Interestingly, human retinal blood vessels showed reduced vasodilation responses at high cholesterol levels [8,9]. However, various anatomical and functional differences exist between retrobulbar and retinal blood vessels, e.g., innervation from autonomic nerve fibers, autoregulation, and endothelium-dependent vasodilation and compensation mechanisms [18,26,70,71,72,73,74]. Hence, differences regarding the susceptibility of the endothelial cells to high cholesterol levels between retrobulbar blood vessels and retinal arterioles are possible.

## 5. Conclusions

In conclusion, our work reveals that endothelial function was impaired in the aorta of ApoE−/− mice by increased oxidative stress possibly via the involvement of NOX1 and NOX2. In contrast, endothelial function was preserved in the ophthalmic artery of ApoE−/− mice, and no changes in endothelial vasodilatory signaling pathways, ROS, and pro-oxidative redox gene levels were observed in this vascular bed. These findings suggest that the mouse ophthalmic artery is resistant to hypercholesterolemia in contrast to other blood vessels. A low expression level or lack of function of LOX-1 may be a possible protective factor against the effects of hypercholesterolemia in the ophthalmic artery. A blockade of the LOX-1 receptor may be an interesting clinical approach to delay functional and morphological vascular alterations in hypercholesterolemia.

## Figures and Tables

**Figure 1 diseases-11-00124-f001:**
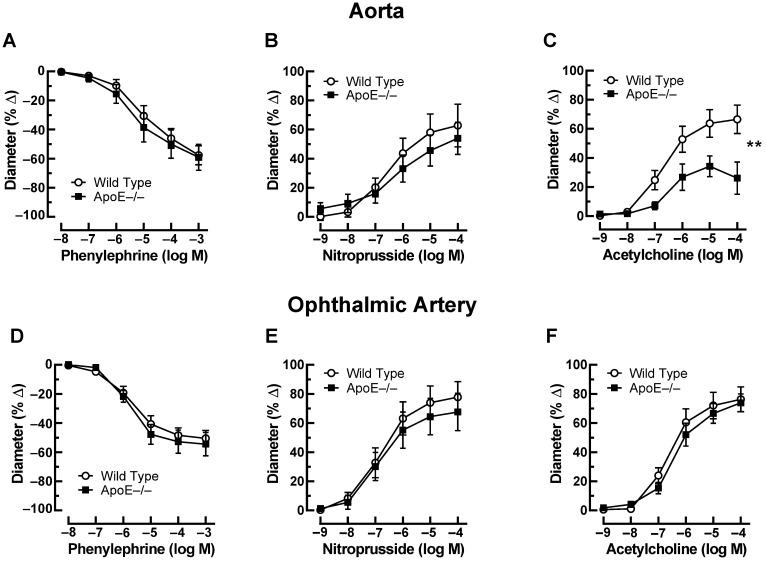
Responses of the aorta from ApoE−/− and wild-type mice to phenylephrine (**A**), to nitroprusside (**B**), and to acetylcholine (**C**). Of note, reactivity of the aorta to acetylcholine was impaired in ApoE−/− mice. Ophthalmic artery reactivity to phenylephrine (**D**), to nitroprusside (**E**), and to acetylcholine (**F**) did not differ between both mouse genotypes. Data are presented as mean ± SE (n = 8 per concentration and genotype; ** *p* < 0.01).

**Figure 2 diseases-11-00124-f002:**
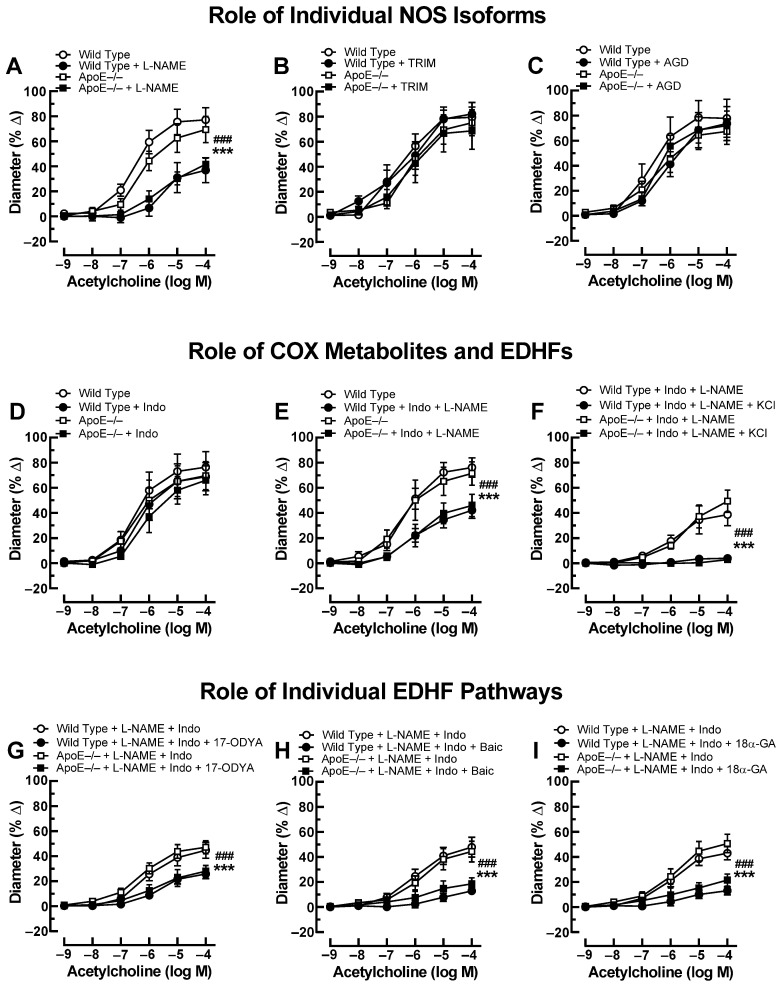
Ophthalmic artery responses of ApoE−/− and wild-type mice to acetylcholine before and after incubation with the non-isoform-selective NOS inhibitor, Nω-nitro-L-arginine methyl ester (L-NAME) (**A**), the selective nNOS blocker, 1-[2-(trifluoromethyl)phenyl]imidazole (TRIM) (**B**), and the selective iNOS inhibitor, aminoguanidine (AGD) (**C**). Among these three NOS blockers, only L-NAME markedly blocked responses in ApoE−/− and wild-type mice, indicative of eNOS involvement. The non-isoform-selective COX inhibitor, indomethacin (Indo) (**D**), had no effect on cholinergic vasodilation, while combined blockade with L-NAME and Indo (**E**) blocked vasodilation responses to a similar extent as L-NAME alone. The residual portion of the vasodilation response in ophthalmic arteries incubated with L-NAME and Indo was almost completely abolished after adding 30 mM of potassium chloride (KCl) (**F**), indicative of endothelium-derived hyperpolarizing factor (EDHF) involvement. Blockade of the cytochrome P450 monoxygenase (CYP450) and 12/15-lipoxygenase pathways with 17-octadecynoic acid (17-ODYA) (**G**) and baicalein (Baic) (**H**), respectively, as well as of gap junctions with 18α-glycyrrhetinic acid (18α-GA) (**I**) reduced vasodilation responses to a similar extent in L-NAME and Indo-incubated ophthalmic arteries of both mouse genotypes, suggesting contribution of these pathways to vasodilation. Data are presented as mean ± SE (n = 8 per concentration and genotype; ### *p* < 0.001, ApoE−/− mice treated versus non-treated; *** *p* < 0.001, wild-type mice treated versus non-treated).

**Figure 3 diseases-11-00124-f003:**
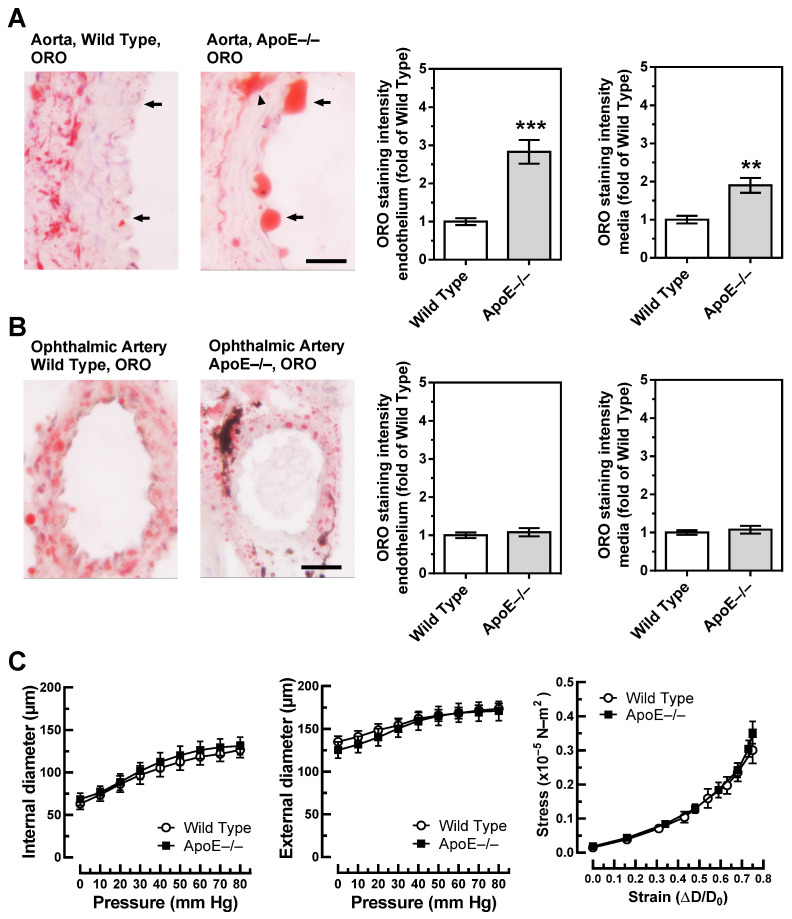
Lipid content, visualized by Oil Red O (ORO, red color) was markedly increased in the aorta of ApoE−/− mice and was especially pronounced in the endothelium (**A**). In contrast, lipid content was similar in the ophthalmic artery from wild-type and ApoE−/− mice (**B**). The dark areas in the ophthalmic artery cross-section of the ApoE−/− mouse are melanin deposits, which are physiologically found in the vascular wall of pigmented mice and are not specific for ApoE−/− mice. Relation between vascular internal and external diameters (µm) to the luminal pressure (mmHg), as well as the stress (in ×10^−5^ N-m^2^)–strain (in ΔD-D₀) relation did not differ between ophthalmic arteries of wild-type and ApoE−/− mice (**C**). Data are presented as mean ± SE; *** *p* < 0.0001; ** *p* < 0.001. The black arrows point to the endothelium, and the arrowhead points to a lipid-rich lesion in the media; scale bar = 20 µm.

**Figure 4 diseases-11-00124-f004:**
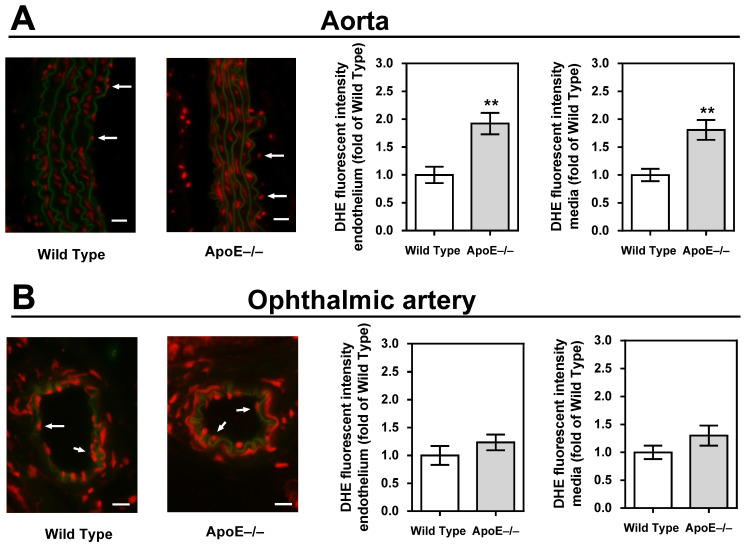
Dihydroethidium (DHE) staining intensity (red) was markedly increased in the endothelium and the media of the aorta (**A**) but not in the ophthalmic artery (**B**). For each vascular bed and mouse genotype, a representative photograph is presented as a merged picture of the DHE staining and the autofluorescence of the elastic lamina (green) to visualize the localization of cells within the vascular wall. The white arrows point to the endothelial layer. Data are presented as mean ± SE normalized to wild-type controls (n = 8 per genotype; ** *p* < 0.01, ApoE−/− versus wild-type mice). Scale bar = 20 µm.

**Figure 5 diseases-11-00124-f005:**
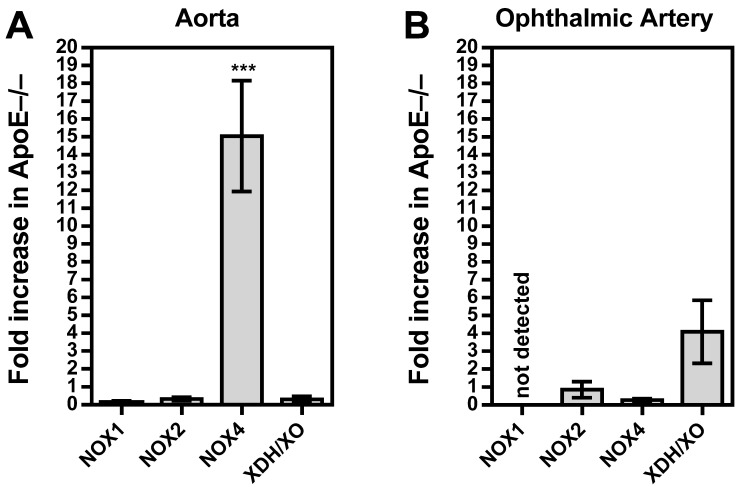
Messenger RNA expression for the pro-oxidative redox enzymes NOX1, NOX2, NOX4, and XDH/XO in the aorta (**A**) and the ophthalmic artery (**B**) presented as the fold change (mean ± SE) in ApoE−/− mice versus wild-type controls (aorta: n = 8 per genotype; ophthalmic artery: n = 6 per genotype; *** *p* < 0.001, ApoE−/− versus wild-type mice).

**Figure 6 diseases-11-00124-f006:**
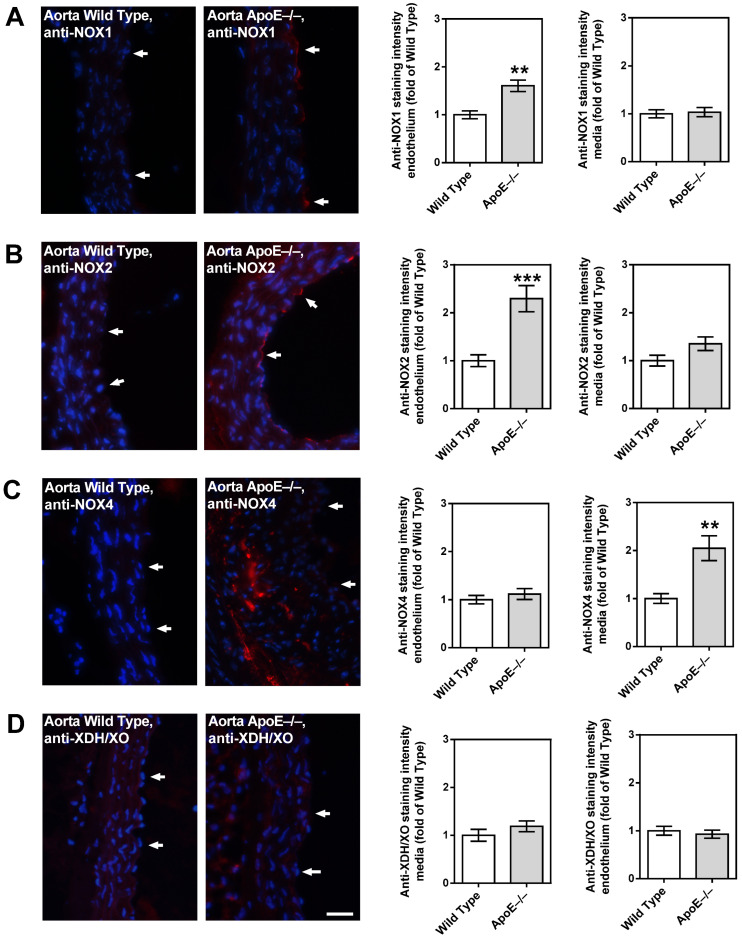
Immunofluorescence staining for pro-oxidative redox enzymes in aortic tissue sections. Immunoreactivity to NOX1 was markedly elevated in the endothelium of ApoE−/− mice (**A**). Likewise, NOX2 immunoreactivity was pronounced in the ApoE−/− mouse endothelium (**B**). Interestingly, NOX4 immunoreactivity was not elevated in the endothelium, but in the media of ApoE−/− mice (**C**). Immunoreactivity to XDH/XO was weak in both wild-type and ApoE−/− mice (**D**). Data are presented as mean ± SE normalized to wild-type controls (n = 8 per genotype; *** *p* < 0.001; ** *p* < 0.01, ApoE−/− versus wild-type mice). The white arrows point to the endothelial layer. Blue: DAPI; Red: Rhodamine Red-X-coupled. Scale bar = 20 µm.

**Figure 7 diseases-11-00124-f007:**
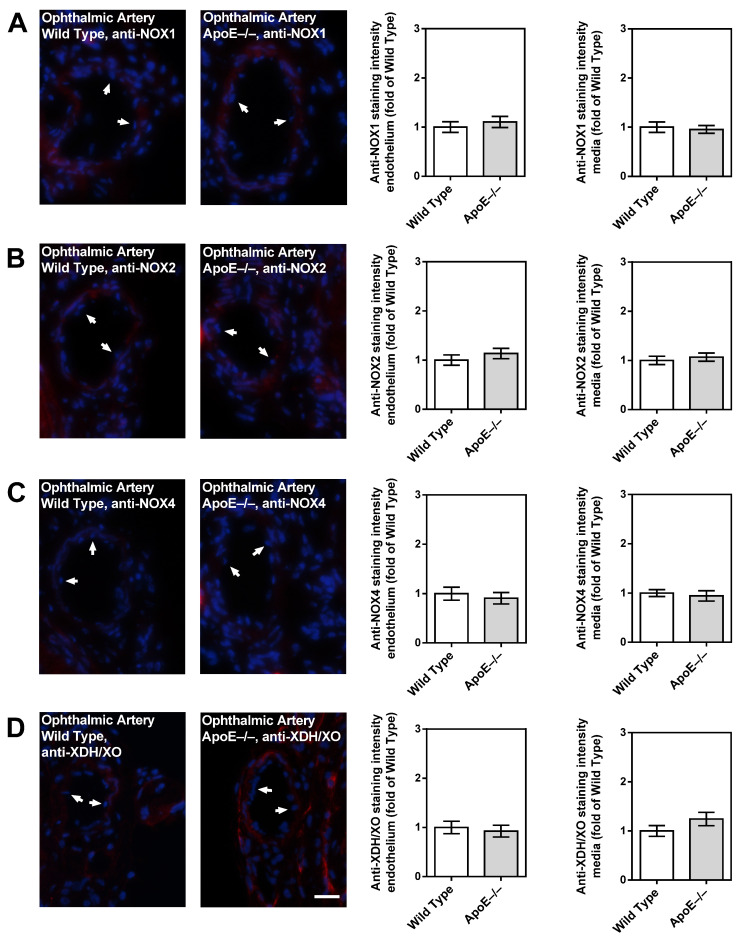
Immunofluorescence staining for pro-oxidative redox enzymes in ophthalmic artery cross-sections. Immunoreactivity to NOX1 (**A**), NOX2 (**B**), and NOX4 (**C**) was negligible in both wild-type and ApoE−/− mice. Only faint immunoreactivity to XDH/XO was visible in the media and adventitia of the ophthalmic artery but was similar in wild-type and ApoE−/− mice (**D**). Data are presented as mean ± SE normalized to wild-type controls (n = 8 per genotype). The white arrows point to the endothelial layer. Blue: DAPI; Red: Rhodamine Red-X-coupled. Scale bar = 20 µm.

**Figure 8 diseases-11-00124-f008:**
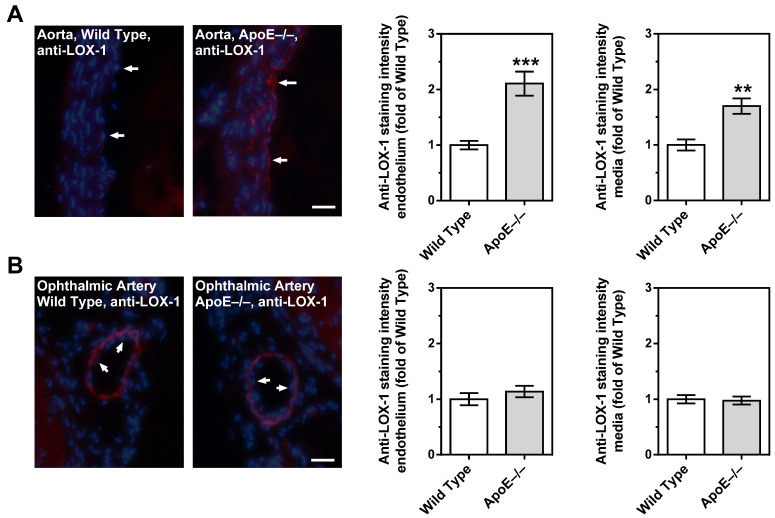
LOX-1 immunoreactivity in the aorta and in the ophthalmic artery of wild-type and ApoE−/− mice. In the aorta from ApoE−/− mice, immunoreactivity to LOX-1 was increased in both the endothelium and the smooth muscle compared to wild-type mice (**A**). In contrast, LOX-1 staining was very faint in the endothelium of the ophthalmic artery from both mouse genotypes (**B**). Although pronounced LOX-1 immunoreactivity was observed in the ophthalmic artery smooth muscle, there was no difference in staining intensity between ApoE−/− and wild-type mice. Data are presented as mean ± SE normalized to wild-type controls (n = 8 per genotype; *** *p* < 0.001; ** *p* < 0.01, ApoE−/− versus wild-type mice). The white arrows point to the endothelial layer. Blue: DAPI; Red: Rhodamine Red-X-coupled. Scale bar = 20 µm.

**Figure 9 diseases-11-00124-f009:**
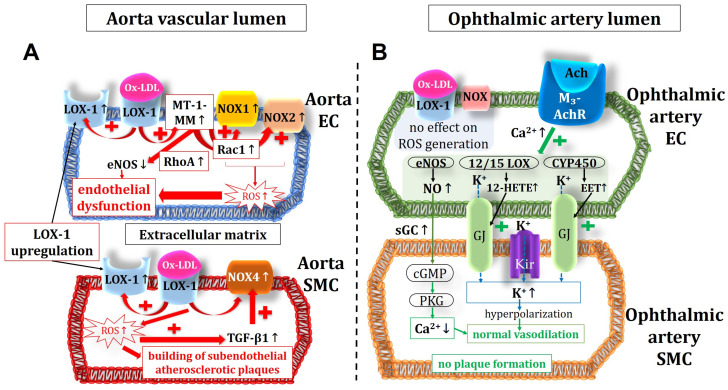
Intracellular pathways in the vascular endothelium and in the smooth muscle of the aorta (**A**) and the ophthalmic artery (**B**) in ApoE−/− mice. Of note, in the aorta, the expression of NOX1 and NOX2, responsible for the ROS production and the endothelial dysfunction, were increased in the endothelium but not in the smooth muscle, where an increased expression and activity of NOX4 could be shown. This suggests that NOX4 plays a role in modification of atherosclerotic plaques in the vascular smooth muscle layer. Unlike in the aorta, in the ophthalmic artery, no LOX-1 upregulation and no increase in ROS production were observed. Moreover, endothelium-dependent vasodilatory mechanisms were not altered in the ophthalmic artery. Ach: acetylcholine; M_3_-AchR: M_3_ muscarinic acetylcholine receptor; EC: endothelial cell; SMC: smooth muscle cell; MT-1-MM: membrane type 1 matrix metalloproteinase; RhoA: Ras homolog family member A; Rac1: Ras-related C3 botulinum toxin substrate 1; Ox-LDL: oxidative low-density lipoprotein; LOX-1: lectin-like oxidized low-density lipoprotein receptor 1; NOX: nicotinamide adenine dinucleotide phosphate oxidase; ROS: reactive oxygen species; TGF-β1: transforming growth factor β1; eNOS: endothelial nitric oxide synthase; Kir: inwardly rectifying potassium channels; 12/15 LOX: 12/15 lipoxygenase; CYP450: cytochrome p450 monoxygenase; NO: nitric oxide; sGC: soluble guanylate cyclase; cGMP: cyclic guanosine monophosphate; PKG: protein kinase G; GJ: gap junctions; EET: epoxyecosatrienoic acid; 12-HETE: 12-hydroxyeicosatetraenoic acid. Up arrows mean increase or upregulation, and down arrows mean decrease or downregulation.

**Table 1 diseases-11-00124-t001:** Primer sequences used for quantitative PCR analysis.

Gene	Accession Number	Forward	Reverse
*NOX1*	NM_172203.2	GGTTGGGGCTGAACATTTTTC	TCGACACACAGGAATCAGGAT
*NOX2*	NM_007807.5	GCACCTGCAGCCTGCCTGAATT	TTGTGTGGATGGCGGTGTGCA
*NOX4*	NM_015760.5	GGCTGGCCAACGAAGGGGTTAA	GAGGCTGCAGTTGAGGTTCAGGACA
*XDH*	NM_011723.3	CGATGACGAGGACAACGGTA	TGAAGGCGGTCATACTTGGAG
*ACTB*	NM_007393.5	CACCCGCGAGCACAGCTTCTTT	AATACAGCCCGGGGAGCATC

**Table 2 diseases-11-00124-t002:** Specifications of primary antibodies used for immunofluorescence studies.

Target Antigen	Catalog Number, Company	Species, Clonality	Dilution
NOX1	ab131088, Abcam, Waltham, MA, USA	Rabbit, polyclonal	1:200
NOX2	ab129068, Abcam, Waltham, MA, USA	Rabbit, monoclonal	1:200
NOX4	ab109225, Abcam, Waltham, MA, USA	Rabbit, monoclonal	1:200
XDH/XO	NBP2-75717, Bio-Techne GmbH, Wiesbaden, Germany	Rabbit, monoclonal	1:50
LOX-1	bs-2044R, BIOZOL Diagnostica Vertrieb GmbH, Eching, Germany	Rabbit, polyclonal	1:100

**Table 3 diseases-11-00124-t003:** Blood pressure and total serum cholesterol in wild-type and ApoE−/− mice. Data are presented as mean ± SE (n = 8 per genotype).

Systemic Parameters	Wild-Type	ApoE−/−	*p* Value
Systolic blood pressure (in mmHg)	103 ± 4.50	112 ± 4.85	0.1678
Diastolic blood pressure (in mmHg)	73.2 ± 3.90	65.0 ± 5.61	0.2521
Mean blood pressure (in mmHg)	82.6 ± 3.89	80.5 ± 4.78	0.7354
Total serum cholesterol (mg/dL)	149 ± 5.79	501 ± 14.9	<0.0001

## Data Availability

The data presented in this study are available on request from the corresponding author.

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
