# Peer review of "Studies on the Effects of Hypercholesterolemia on Mouse Ophthalmic Artery Reactivity"

_diseases, 2023, doi:10.3390/diseases11040124_

Round 1
Reviewer 1 Report
Authors: Francesco Buonfiglio et al.
Title: Effects of Chronic Apolipoprotien E Deficiency on Mouse Ophthalmic Artery Reactivity
Comments: This is a well-written and carefully executed studies on investigating the potential effects of hypercholesterolemia on Ophthalmic artery in mice.
The comments are as follows:
1. Since there were no obvious effects on this study of Ophthalmic artery in Apoe-/- mice, the better title would be “Studies of the effect of hyperlipidemia on mouse ophthalmic artery reactivity”. Otherwise, the original title is misleading the readers.
2. In Fig 3B, please explain the darker stain on the ophthalmic artery of ApoE-/- ORD staining.
3. In Fig. 4, please explain those arrows point to what?
4. In Fig. 8B, both ApoE-/- and wild type mice have distinct, strong positive pink staining on the media (smooth muscle) but not in the endothelium? Is that correct? They are strong staining, not “very low” as stated. Please correct the description.
5. Line 482, the written was “double negative”. It would be better to say “eNOS was function normal” in ophthalmic artery of Apoe-/- mice.
Author Response
We thank the Reviewer for the comments and suggestions. According to these suggestions, we made changes in the text.
1.) Since there were no obvious effects on this study of Ophthalmic artery in ApoE-/- mice, the better title would be “Studies of the effect of hyperlipidemia on mouse ophthalmic artery reactivity”. Otherwise, the original title is misleading the readers.
Response to 1.) Thank you for the suggestion. We changed the title of the article accordingly.
2.) In Fig 3B, please explain the darker stain on the ophthalmic artery of ApoE-/- ORD staining.
Response to 2.) The dark stain is melanin pigment, which is physiologically found along the ophthalmic artery wall in pigmented mice, such as C57BL/6J mice. The pigmented parts were excluded from densitometric analysis, since they had a high degree of variability. We added a text passage on this point in the Materials and Methods section (lines 174–177) and in the figure legend (lines 330–332).
3.) In Fig. 4, please explain those arrows point to what?
Response to 3.) In Figure 4, the white arrows point to the endothelial layer. We added a passage in the figure legend (line 348).
4.) In Fig. 8B, both ApoE-/- and wild type mice have distinct, strong positive pink staining on the media (smooth muscle) but not in the endothelium? Is that correct? They are strong staining, not “very low” as stated. Please correct the description.
Response to 4.) In Fig.8B, LOX-1 expression was very low in both endothelium and smooth muscle of ApoE-/- mice when compared to their aorta sections. However, we agree with the Reviewer in terms of immunoreactivity in the smooth muscle layer and changed the methods section 3.7. and the legend of Figure 8 accordingly (lines 398–412).
5.) Line 482, the written was “double negative”. It would be better to say “eNOS was function normal” in ophthalmic artery of ApoE-/- mice.
Response to 5.) We changed the sentence as suggested to: ”eNOS function was normal in the ophthalmic artery of ApoE-/- mice” (line 477).
Reviewer 2 Report
In the submitted study Buonfiglio and colleagues provide evidence indicating than LOX1- and NOX-4-induced ROS generation in SMCs contributes to endothelial dysfunction in aorta but not in ophthalmic artery. This is an interesting and novel observation, which may be of interest to readership of “Diseases”. The study design is clearcut, the methods used in the study are sophisticated and skilfully applied, and the manuscript is generally well-written. However, there still few points requiring further attention of the authors:
1.) In the manuscript the authors on several occasions underscore the potentially pivotal role of oxLDL as a causal factor precipitating oxidative stress in arterial wall, but the supporting experimental evidence is missing. The authors should attempt to analyse the lipoprotein profile in plasma from mice examined in the study, determine LDL and LDLox concentrations and – if possible - assess the distribution of LDLox epitopes in the intima and media of aortas and ophthalmic arteries.
2.) The study would much profit, if the authors could recapitulate at least some mechanistic pathways revealed in the study under in vitro conditions. For instance, they might consider assessing the LOX1-mediated effect of oxLDL on NOX-4 expression and ROS production in SMCs.
3.) The present Discussion section of the study could be shortened by 20 – 30%. On the other hand, the authors should more extensively address the major point of the study that is why ophthalmic artery and aorta so dramatically differ in their response to LDLox and what might be alternative mechanisms affecting the function of ophthalmic artery in course of hypercholesterolemia.
Minor comments:
The axes and figure descriptions use very small font size, which could probably be used for the examination of visual impairment but is hardly legible for the normal reader. In fig. 8, upper left panels axes descriptions should be corrected.
No specific comments
Author Response
We thank the reviewer for the comments and suggestions. According to these suggestions, we made changes in the text.
1.) In the manuscript the authors on several occasions underscore the potentially pivotal role of oxLDL as a causal factor precipitating oxidative stress in arterial wall, but the supporting experimental evidence is missing. The authors should attempt to analyze the lipoprotein profile in plasma from mice examined in the study, determine LDL and oxLDL concentrations and – if possible - assess the distribution of oxLDL epitopes in the intima and media of aortas and ophthalmic arteries.
Response to 1.) Previous investigations already reported on elevated LDL and Ox-LDL plasma levels in ApoE-/- mice under both standard chow and atherogenic diet conditions (Chang et al. 2023; Hong et al. 2015; Kato et al. 2009; Staprans et al. 2000). Moreover, the increase in Ox-LDL levels has been linked to the progression of atherosclerotic lesions in ApoE-/- mice (Kato et al. 2009). Because ApoE-/- mice have already been well characterized regarding their lipid profile, we decided not to dive deeper into this topic. However, we added a passage on this issue and five additional citations in the Discussion (lines 430–434).
2.) The study would much profit, if the authors could recapitulate at least some mechanistic pathways revealed in the study under in vitro conditions. For instance, they might consider assessing the LOX1-mediated effect of oxLDL on NOX-4 expression and ROS production in SMCs.
Response to 2.) We added a passage on the mechanistic pathways concerning ROS and NOX4 (lines 505–509). Moreover, the effects of Ox-LDL on LOX1-mediated effects are discussed in more detail (lines 517–526).
3.) The present Discussion section of the study could be shortened by 20 – 30%. On the other hand, the authors should more extensively address the major point of the study that is why ophthalmic artery and aorta so dramatically differ in their response to oxLDL and what might be alternative mechanisms affecting the function of ophthalmic artery in course of hypercholesterolemia.
Response to 3.) We shortened the Discussion where possible. Moreover, we added a passage underscoring, why we think LOX-1 may be important in inducing endothelial dysfunction and morphological changes in some blood vessels (lines 526–530). We also explicitly address the point, why the aorta and ophthalmic artery may differ in response to hypercholesterolemia (lines 535–539). This point is also addressed in fig. 9 (lines 544–551).
4.) Minor comments:
The axes and figure descriptions use very small font size, which could probably be used for the examination of visual impairment but is hardly legible for the normal reader. In fig. 8, upper left panels axis descriptions should be corrected.
Response to 4.) We increased the font size, where possible. We also corrected the axis descriptions in fig. 8.
Chang GR, Cheng WY, Fan HC, Chen HL, Lan YW, Chen MS, Yen CC, Chen CM (2023) Kefir peptides attenuate atherosclerotic vascular calcification and osteoporosis in atherogenic diet-fed ApoE (-/-) knockout mice. Front Cell Dev Biol 11: 1158812. doi: 10.3389/fcell.2023.1158812
Hong W, Xu XY, Qiu ZH, Gao JJ, Wei ZY, Zhen L, Zhang XL, Ye ZB (2015) Sirt1 is involved in decreased bone formation in aged apolipoprotein E-deficient mice. Acta Pharmacol Sin 36: 1487-96. doi: 10.1038/aps.2015.95
Kato R, Mori C, Kitazato K, Arata S, Obama T, Mori M, Takahashi K, Aiuchi T, Takano T, Itabe H (2009) Transient increase in plasma oxidized LDL during the progression of atherosclerosis in apolipoprotein E knockout mice. Arterioscler Thromb Vasc Biol 29: 33-9. doi: 10.1161/atvbaha.108.164723
Staprans I, Pan XM, Rapp JH, Grunfeld C, Feingold KR (2000) Oxidized cholesterol in the diet accelerates the development of atherosclerosis in LDL receptor- and apolipoprotein E-deficient mice. Arterioscler Thromb Vasc Biol 20: 708-14. doi: 10.1161/01.atv.20.3.708
Round 2
Reviewer 2 Report
No further comments.
No specific comments